# Utilising eDNA Methods and Interactive Data Dashboards for Managing Sustainable Drinking Water

Sophie E. Watson [1,*], Charlotte H. Taylor [2], Veronica Bell [1], Annalise S. Hooper [1], Thomas R. Bellamy [2], Peter Kille [2] and Rupert G. Perkins [1]

1   School of Earth and Environmental Science, Cardiff University, Main Building, Museum Avenue, Cardiff CF10 3AX, UK; bellv2@cardiff.ac.uk (V.B.); hooperas@cardiff.ac.uk (A.S.H.); perkinsr@cardiff.ac.uk (R.G.P.)
2   School of Bioscience, Cardiff University, Sir Martin Evans Building, Museum Avenue, Cardiff CF10 3AX, UK; taylorc22@cardiff.ac.uk (C.H.T.); bellamyt@cardiff.ac.uk (T.R.B.); kille@cardiff.ac.uk (P.K.)
*   Correspondence: watsons2@cardiff.ac.uk

**Abstract:** Generating rapid, easy-to-interpret community data for drinking reservoirs as a means of tackling water quality management is of increasing demand within the water industry. Taste and odour (T&O) is one of many increasing concerns to water companies worldwide, incurring huge costs as customer complaints accumulate and additional treatment and resource management are required. However, there remains a two-fold issue in addressing T&O management: firstly, predicting the initial onset of a T&O event relies on a highly complex understanding of environmental considerations and their interaction with T&O-related taxa, and secondly, there remains a lag between the notification of a T&O event and the resolution of the issue by reservoir management staff. This is partly due to slow, low-resolution methods of detecting and reliably identifying problem taxa in samples. These methods are unable to provide information on the huge plethora of taxa related to T&O metabolite production and often cannot provide data in a timely enough manner for an opportune management response. This means the water industry is often forced to use a reactive, rather than proactive, approach to water quality monitoring. Here, we present methods for implementing a high-throughput sequencing approach to monitoring drinking reservoirs for water quality and improving the sustainability of water supplies, as well as methods for presenting these data on easy-to-interpret data dashboards that can be updated rapidly as new data are generated. Our methods and dashboarding approaches are currently being trialled and tested within the UK water industry, and so here, we show anonymised examples of those data presentations. We propose that these methods can greatly aid reservoir management teams in their approach to T&O monitoring and can be used to implore more sustainable management pipelines, safeguarding future water sources.

**Keywords:** reservoir management; sustainable water sources; data dashboarding; taste and odour (T&O); eDNA; community analysis

## 1. Introduction

Only 2% of the Earth's water sources are made up of freshwater, with it being predicted that freshwater resources could become the most critical factor in the future of our planet [1–3]. Globally, freshwater supplies are being depleted at a far faster rate than they can be regenerated by rainfall [2–4] and are increasingly exposed to extreme weather events and pollution influx from various anthropogenic sources [5]. Shifts in catchment land use and climatic conditions can additionally lead to the erosion of ecological stability within freshwater systems, removing feedback loops required to maintain ecological resilience. In the absence of ecological equilibrium, processes that would otherwise buffer the decline in water quality no longer exist. It is through this process that the establishment of harmful algal blooms (HABs) occur in freshwater environments, threatening the sustainability of drinking water supplies.

HABs, which predominantly consist of blue-green algae (correctly named Cyanobacteria) [6], are a frequent concern to water quality and therefore to the sustainability of drinking water supplies. Left untreated, these blooms can result in filter blocking at water treatment plants or treatment breakthrough and, as such, are a significant and costly problem for water companies [7]. In addition, algal blooms are often associated with taxa concurrent to water quality risks, including toxin and taste and odour (T&O) metabolite production.

The predominant method by which consumers assess the quality of their drinking water is by its palatability, including the aesthetic aspects of taste and odour quality. The occurrence of unpleasant tastes and odours in drinking water is becoming increasingly prevalent worldwide, an issue which is most commonly associated with two volatile compounds, geosmin (trans-1,10-dimethyl-trans-9 decalol-$C_{12}H_{22}O$) and 2-methylisoborneol (2-methyl isoborneol-$C_{11}H_{20}O$; 2-MIB). Both geosmin and 2-MIB have extremely low human detection thresholds (between 4 and 10 ng/L) [8]. The production of geosmin and 2-MIB in drinking water reservoirs is predominantly linked to members of the bacterial phylum Cyanobacteria [9]. Production of the compounds occurs during rapid growth when conditions are optimal [10,11]. Phytoplankton community diversity and abundance are linked to a complex array of environmental variables, including reduced summer rainfall and increased average temperatures [11,12], as well as nutrient influx into reservoir systems. Ammonium and inorganic phosphorous from agricultural practices and fertilizer run-off can exacerbate T&O issues [10], while extreme weather events, e.g., excessive summer rainfall and flooding, can lead to unpredictable pulses of nutrients into the water catchment, exacerbating T&O metabolite production.

Despite there being no human health risk from the ingestion of geosmin and 2-MIB [13–15], water companies are under increasing pressure to address complaints relating to their levels in drinking water. Treatment breakthrough and customer complaints relating to T&O events incur considerable costs to water companies, an issue which is likely to worsen in the future due to increased climatic events and agricultural pressures [16–18]. According to a number of UK water companies, increased treatment requirements and customer complaints in response to T&O events are predicted to cost over GBP 200 million per year. Between 2002 and 2012, one US city (inhabited by ~165,000 residents) estimated that T&O issues cost them around USD 70.4 million in treatments and a further USD 6.9 to 10.3 million in lost revenue [19].

A number of different treatment methods are currently utilised within the water industry as a means of utilising the impact of algal blooms and T&O events, for example, coagulation, sedimentation, filtration, chlorination, activated carbon or oxidation such as ozone [20,21]. However, these treatments can be costly, and understanding the precise timeframe in which to execute a given treatment can be incredibly complex. What is needed is a way of monitoring shifts in phytoplankton communities across water catchment and reservoir systems in near real time in relation to environmental variables. Phytoplankton community patterns can provide a potential method for evaluating ecological responses to stressors and, as such, potential water quality risks. As such, here we propose a molecular method for rapidly monitoring the magnitude of different bacterial taxa across different reservoir sites simultaneously, as well as a method for presenting these data alongside other variables of interest (e.g., nutrient and metabolite concentrations) in easy-to-interpret dashboarding systems.

Molecular methods have the potential to provide a clearer understanding of how Cyanobacterial communities respond to multiple variables, across numerous sites, in rapid time frames. Currently, a lot of industry practice relies on cell count and microscopy methods in order to categorise Cyanobacterial diversity and abundance [7], which can be slow and may not provide the taxonomic resolution required for optimal management decisions. Instead, amplicon targeted sequencing can be used to target genes that occur ubiquitously across a taxonomic group of interest, for example, the 16S rRNA gene in bacteria [22]. These genes harbour a huge amount of hypervariability, which enables accurate species-level distinctions of hundreds, if not thousands, of taxa present within

each sample [22]. As such, this provides a reliable method for capturing the total diversity (and therefore shifts in diversity, abundance or community structure) of Cyanobacteria within freshwater. Due to its rapid, high-throughput approach, amplicon sequencing has become an integral and widely utilized method for inferring the presence of taxonomic groups [22] but is yet to be adopted as part of long-term monitoring by the water industry.

Understanding and interpreting sequencing data, alongside large metadata data frames, can be a daunting endeavour for those not trained in bioinformatics and is therefore of limited use to reservoir managers and water industry staff. However, here we present ways in which these data can be presented as interactive data dashboards, with simple data refreshing capabilities and therefore rapid data visualization turnaround times. We present examples of how we have implemented these methods as a means of aiding reservoir management decisions within the UK water industry, thus aiding the sustainability and safeguarding of water sources. Currently within the UK, and indeed globally, amplicon targeted sequencing is not yet commonly utilized by the water industry as a method of monitoring water quality, but the implementation of such methods could be transformative for reservoir management. The aim of this manuscript is to facilitate the implementation of molecular data in reservoir management with the objective of presenting a dashboarding system that can be used to evidence base shifts in water quality risk.

## 2. Materials and Methods

### 2.1. Sampling Sites

A total of 29 eDNA samples were collected from three sampling points along a UK reservoir between June and December 2022. The name and location of the reservoir have been utilized for the purpose of data protection and have been provided as an example of how data can be utilized and presented. In brief, at each sampling point, 500 mL of reservoir water was filtered across a 0.45 μm self-preserving flat filter [23] using a Smith-Root Citizen Science eDNA Sampler (Smith-Root, Inc.©, Vancouver, WA, USA). Filters were stored at Cardiff University inside their sampler casing at room temperature until later use, as per manufacturer instructions [23]. In parallel, a 500 mL water sample was collected from each site in a sterile bottle and refrigerated at 4 °C until later water chemistry analysis.

### 2.2. DNA Extraction from eDNA Filters

Each eDNA filter was cut in half; one half of the filter was stored for later use and the other half was used within the DNA extraction process. Using sterile techniques, the filter half was cut into small sections and transferred to a sterile 15 mL falcon tube. A total of 460 μL of ATL Buffer (Qiagen Ltd., Manchester, UK) and 40 μL of Proteinase K were added to each sample, before briefly vortexing and centrifuging. Samples were then incubated for 30 min in a water bath at 56 °C. Each sample was then subjected to a freeze–thaw process: 10 s in liquid nitrogen followed by incubation within a water bath at 56 °C for 1 min (repeated once before allowing samples to cool on ice for 2 min). Subsequently, 0.4 mL of 0.1 mm and 1.0 mm glass beads (Thistle Scientific Ltd., Rugby, UK) was added to each tube along with 800 μL of extraction buffer (5 M NaCl, 30 mM NaEDTA and 70 mM tris, pH 8.0), 800 μL of chloroform and 100 μL of DTAB (10%). Samples were then rigorously agitated using a bead beater at 5 m/s for 30 s, with 5 min rest (repeated once). Samples were then centrifuged for 10 min at 4250 rpm before adding equal volumes of Buffer AL (Qiagen) and 100% ethanol to the volume utilized. All liquid was transferred to a DNA mini spin column (Qiagen) in 600 μL aliquots and centrifuged for 1 min at 14,000 rpm each time. The remaining DNA extraction steps followed those outlined in the DNeasy Blood and Tissue Kit Protocol for Purification of Total DNA from Animal Blood or Cells (Spin-Column Protocol) (Qiagen; following manufacturer instructions), starting from Step 4 in this commercially available protocol. Extraction controls were run alongside each extraction set.

### 2.3. Amplification and Illumina Sequencing

An approximately 350 base pair (bp) fragment of the 16S rRNA gene was amplified using the bacteria-specific primer set 515F (forward) 5′-GTGCCAGCMGCCGCGGTAA-3′ and 806R (reverse) 5′-GGACTACHVGGGTWTCTAAT-3′ in the following conditions on an Applied Biosystems SimpliAmp thermocycler: 95 °C for 2 min (one cycle), 95 °C for 5 s, 55 °C for 15 s, 72 °C for 10 s (30 cycles) and 72 °C for 5 min (one cycle). Polymerase chain reaction (PCR) was conducted in triplicate to account for bias and included a negative PCR control. Visualisation and quality checking of PCR products were conducted using a QIAxcel® Advanced System before samples were pooled in triplicate and purified using a Zymo Research DNA Clean-up Kit™. Illumina® Nextera XT indices (Illumina®, Cambridge, UK) were incorporated via a secondary PCR using the following steps: 95 °C for 3 min (one cycle), 95 °C for 30 s, 55 °C for 30 s, 72 °C for 30 s (8 cycles) and 72 °C for 5 s (one cycle). Samples were utilized to equimolar concentrations using a SequalPrep™ Normalization Plate Kit (Invitrogen, Carlsbad, CA, USA) before sequencing on an lllumina® MiSeq (2 × 300 bp reads) at Cardiff School of Bioscience Genome Research Hub.

### 2.4. Bioinformatic Analysis of Sequencing Data

Bioinformatic analyses were conducted in QIIME2 2021.8 [24]. Paired-end reads were joined using VSEARCH 2021.8.0 [25] and quality filtered using q-score-joined parameters [26]. Reads were denoised and dereplicated using DADA2 2021.8.0 [27]. Taxonomic assignments of representative sequences from each amplicon sequence variant (ASV) were performed using the SILVA 138 classifier, with 99% similarity. Taxonomic assignment was conducted using Scikit-learn version 0.21 [28]. Subsequent filtering, including the removal of singletons and rarefaction, was performed in the software 'R' (version 4.2.2) [29].

### 2.5. Water Chemistry Analysis

Dissolved inorganic nutrients (orthophosphate (OP) and total phosphorus (TP)) were determined using colourimetric methods described in *Standard Methods for the Examination of Water and Wastewater, 22nd edition* [30]. The method for measuring total nitrogen (TN) follows the digestion mixture protocol detailed in De Borba et al. [31], with the exception that we used an Enzo Life Sciences, Inc. Nitric Oxide detection kit. Ammonium, geosmin and MIB data were provided by the water company whose data are presented within this paper and were completed in a UKAS-accredited laboratory. Ammonium was measured using Hach LANGE GMBH® LCK 303 (DOC312.53.94009), while geosmin and MIB were measured using solid-phase microextraction (SPME) and gas chromatography–mass spectrometry (GCMS) analysis.

### 2.6. Generating Data Dashboards

Tableau Desktop 2022.4.1© was used to generate interactive data dashboards that were then securely published on Tableau Server 2023.2.0©. Tableau software requires licenses both for the creator and the users who intend to view the data. Data that are deemed suitable for public view can be viewed for free on Tableau Public©.

## 3. Results

Monitoring and management of T&O risk in drinking water reservoirs requires accurate, timely data on the community of bacteria. T&O metabolites, geosmin and MIB are predominantly produced by a key set of Cyanobacteria, and assessing their presence and change in abundance requires expensive and time-consuming microscopy by skilled operators in water service labs. Interpretation of analyses is problematic due to the variability between even trained taxonomists, especially for Cyanobacteria. This leads to inaccuracies and delays in relaying the required data to decision makers regarding treatment utilized and preventative intervention management. Using an amplicon targeted approach greatly increases the speed and accuracy with which large numbers of bacterial taxa can be identified. Here, using this approach, 38 bacterial phyla constituting 500 dif-

ferent genera were detected across 29 different samples. In total, six bacterial phyla made up 96% of the total reads (Proteobacteria: 19,226 reads, Planctomycetota: 14,174 reads, Bacteroidota: 13,394 reads, Actinobacteriota: 8451 reads and Cyanobacteria: 6621 reads, making up 69,896 reads total). A number of Cyanobacterial taxa were detected that are known to play an important role in T&O production and water quality risk: Microcystis, Snowella, Planktothrix, Aphanizomenon and Pseudanabaena [8,9]. Additionally, the Cyanobacterium Cyanobium was detected. It has previously been reported that there is a discrepancy between the identification of Cyanobium when using molecular methods compared to microscopy and that, instead, members of the Aphanothece genus are likely actually members of Cyanobium [32]. Aphanothece (or Cyanobium) are typically found in high abundance in nutrient-rich systems [32] and so are also notable taxa when considering water quality metrics of interest.

Using data dashboards, the read abundance of all bacterial phyla present per sample can be presented as stacked bar charts (Figure 1), and data can be utilized by taxa of interest, in this case, Cyanobacteria, plotted at the genus level (Figure 1). Shifts in bacterial diversity per sampling location over time can also be monitored (Figure 2), a factor which can be utilized as an important indicator of instability within an ecosystem [33]. In the absence of dominance in the community, there is little control to suppress the growth of potential problem taxa, but on the other hand, rapid declines in diversity can indicate the increasing abundance or dominance of potentially problematic taxa [33]. As such, a change in diversity can be beneficial to increase resilience to change or alternatively facilitate a dominance of harmful taxa that can result in an increased level of water quality risk such as taste and odour metabolite production by Cyanobacteria. We propose it is useful to monitor shifts in diversity as an early indicator system for broader ecological change, but diversity shifts (in either direction) must be assessed within an ecological and community data context, paying close attention to various relevant environmental and chemical parameters.

A key chemical parameter of interest is phosphate, which has been shown to control phytoplankton growth [34]. Although traditionally research has considered inorganic phosphorous and nitrogen the predominant nutrients controlling algal communities, more recent research has demonstrated that organic phosphorous and nitrogen may be critical nutrient sources too [35]. Phosphorus availability is required for rapid growth in phytoplankton, but monitoring sources and concentrations of ammonium is also considered key for managing T&O outbreaks in drinking water reservoirs [10]. Perkins et al. [10] found that an 85% decrease in ammonium from the primary external loading source was associated with reduced T&O production. The ratio of ammonium to nitrate has also been shown to play a central role in metabolite production, whereby high ratios were associated with an elevated abundance of geosmin production [11]. This may be explained by the fact that Cyanobacteria along with other phytoplankton are more efficient at assimilating ammonium compared to nitrate [36]. On dashboards, such nutrient concentrations can be displayed as tables ordered by reservoir sampling site (Figure 3) and can include nutrient fractions, e.g., OP to TP, which can additionally be coloured using a heatmap (e.g., where lower ratios are in a lighter colour, while higher ratios are in a darker colour) (Figure 3). Alternatively, nutrient fractions, such as OP to TP, TP to TN and ammonium to nitrate ratios, investigated over time series data can be presented as line plots, whereby each line is coloured by sampling site (Figure 4). It is possible to arrange multiple plots on one dashboard in order to identify trends across nutrient fractions (Figure 4). The same visualizations can be used for any other water chemistry parameters of interest, including taste and odour metabolite concentrations (Figure 5), as well as environmental variables, such as temperature, precipitation level and reservoir depth, all of which have been linked to T&O events [11,12,37].

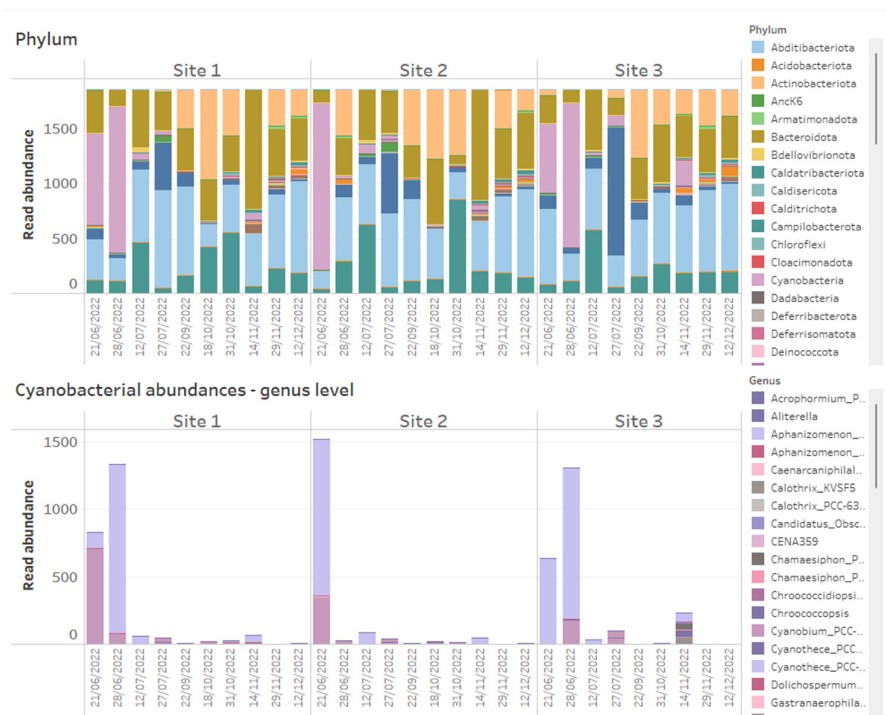

**Figure 1.** Example of how 16S rRNA-targeted high-throughput sequencing data from time series data points of three reservoir sites can be presented on dashboards. Above shows stacked bar charts of the relative abundance of all bacterial phyla detected within time series samples across three reservoir sites, while below are stacked bar charts of absolute abundances of Cyanobacteria, plotted at genus level across the same samples.

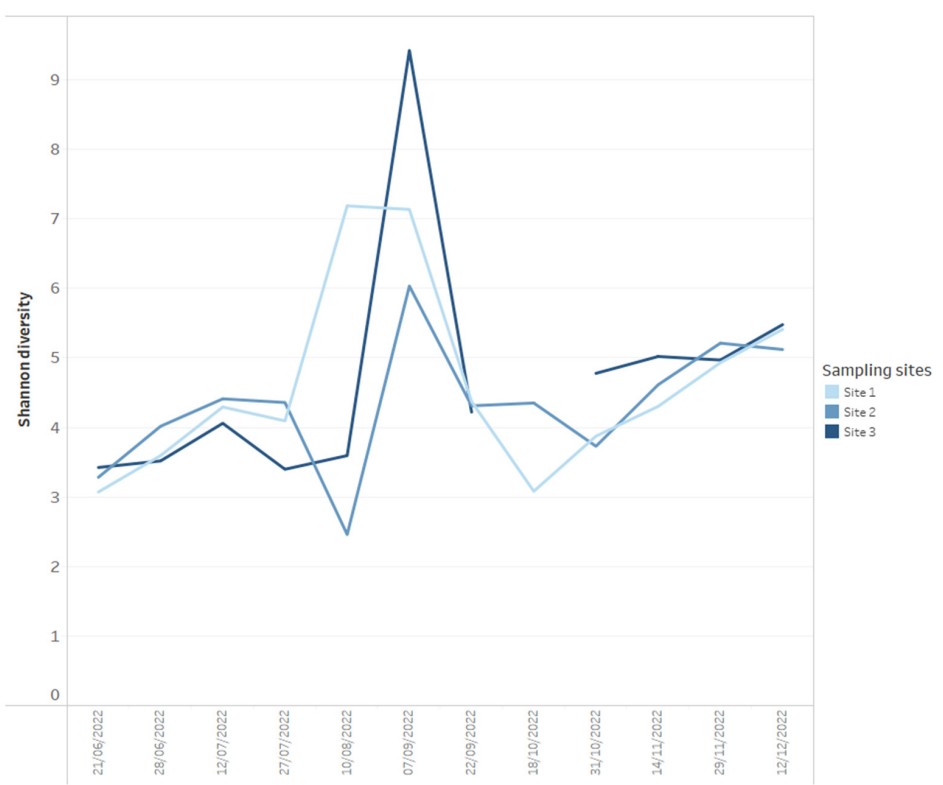

**Figure 2.** Example of how Shannon diversity measures for each time series data point for three reservoir sites (calculated from 16S rRNA-targeted high-throughput sequencing data) can be presented on dashboards.

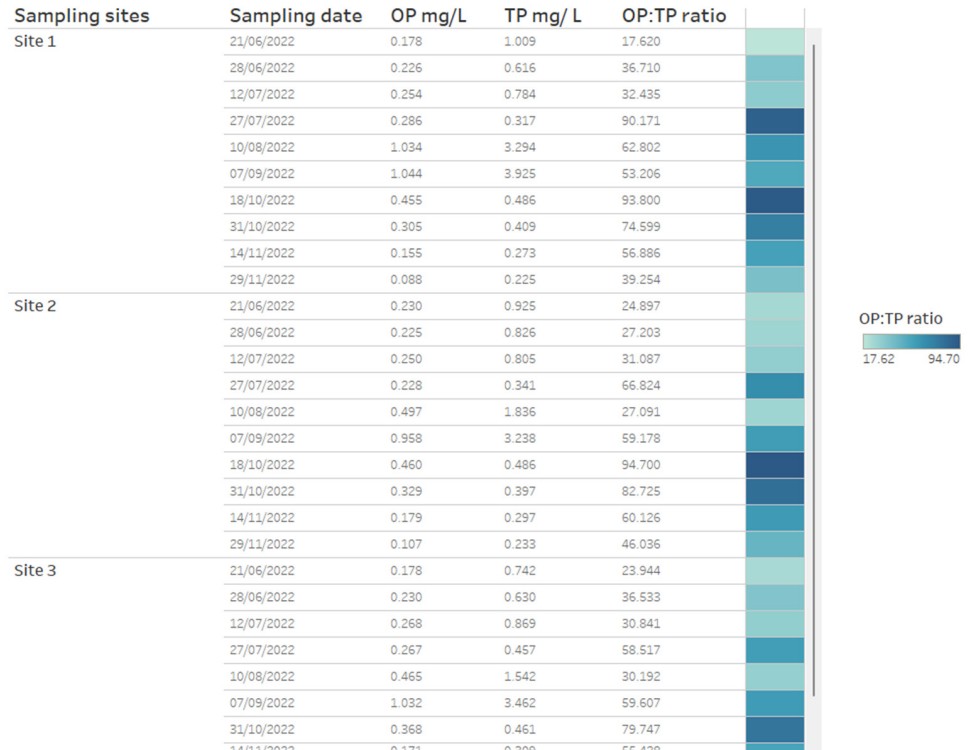

**Figure 3.** Example of how water chemistry analysis for three different reservoir sites can be presented as a table in a dashboarding format. On the right, cells are coloured as a heatmap (light to dark) depending on the orthophosphate (OP) to total phosphate (TP) ratio.

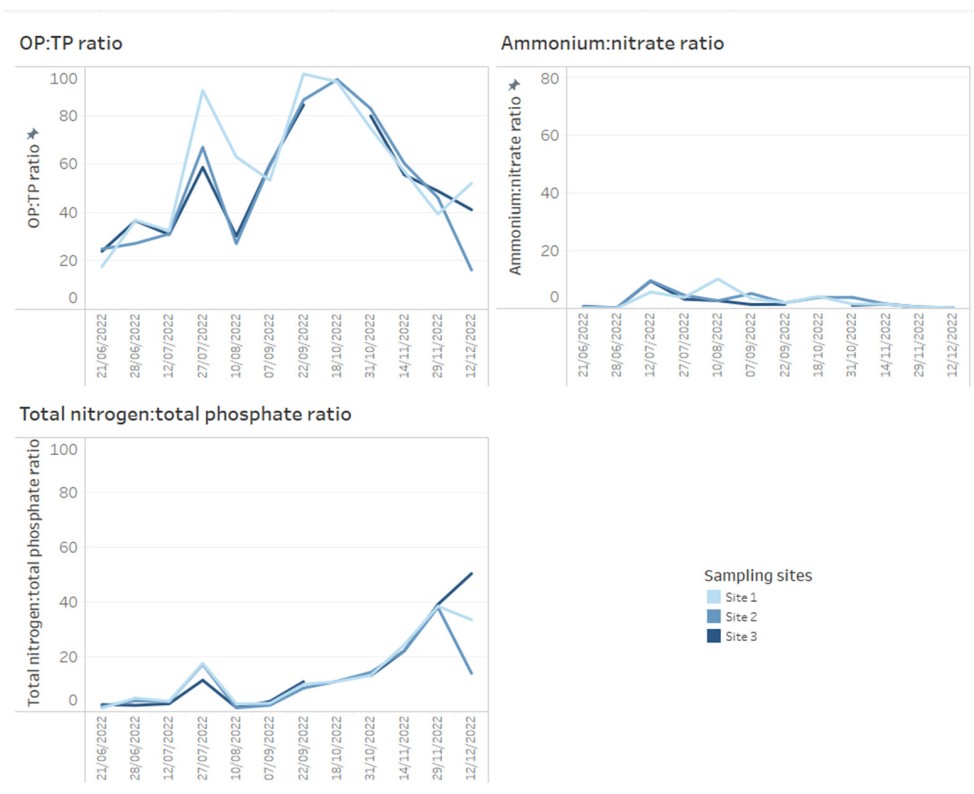

**Figure 4.** Example of how time series data for water chemistry ratios across three reservoir sites can be presented on dashboards. Here, orthophosphate (OP) to total phosphate (TP ratio), ammonium to nitrate and total nitrogen to total phosphate ratios are presented.

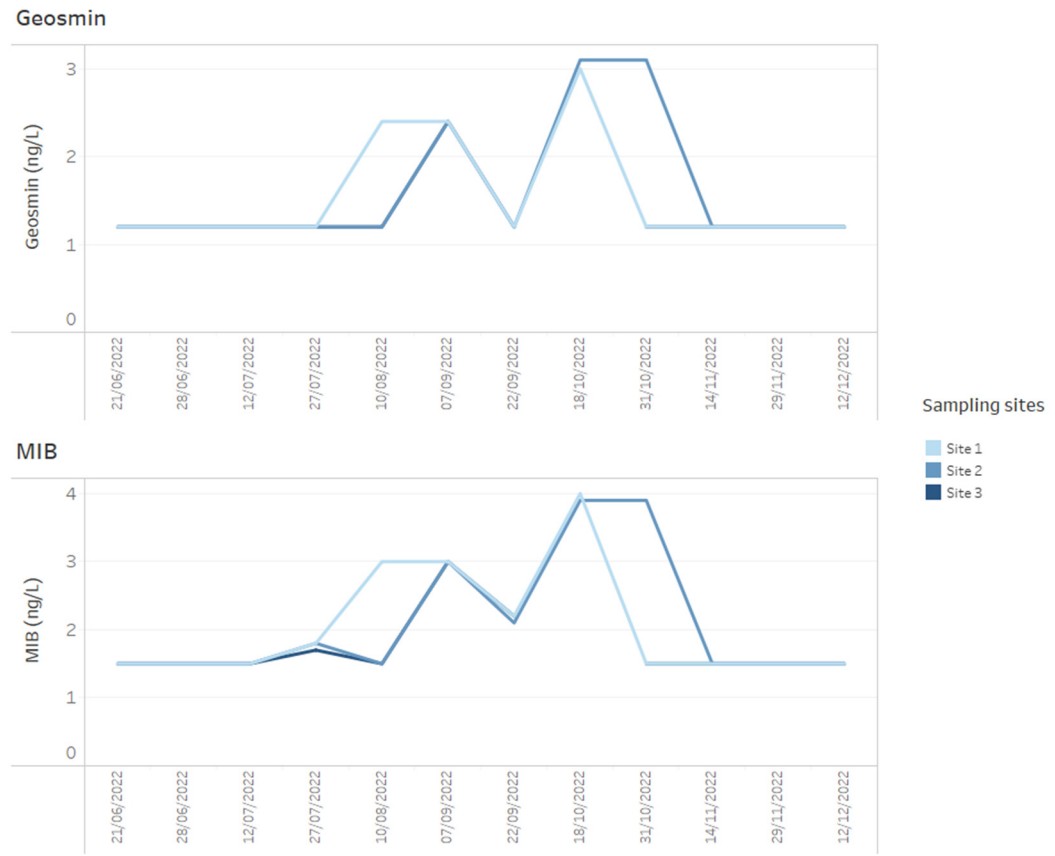

**Figure 5.** Example of how time series data for T&O metabolites, geosmin and MIB for three reservoir sites can be presented on dashboards.

Appreciating the source or extent of a T&O event requires an understanding of how communities change in response to interactions between environmental variables (e.g., temperature, rainfall, reservoir depth) and nutrients (e.g., inorganic and organic nitrogen and phosphorus fractions). Any variables of choice can be included on dashboards or as part of a workbook tab system for easy navigation and comparison (Figure 6). Dashboards are interactive, meaning data can be easily divided into subsets by, for example, sampling sites or taxa of interest (Figure 7), and users can hover over data points to reveal additional information rapidly. These adapted dashboards can then be downloaded and shared with colleagues or saved for future use.

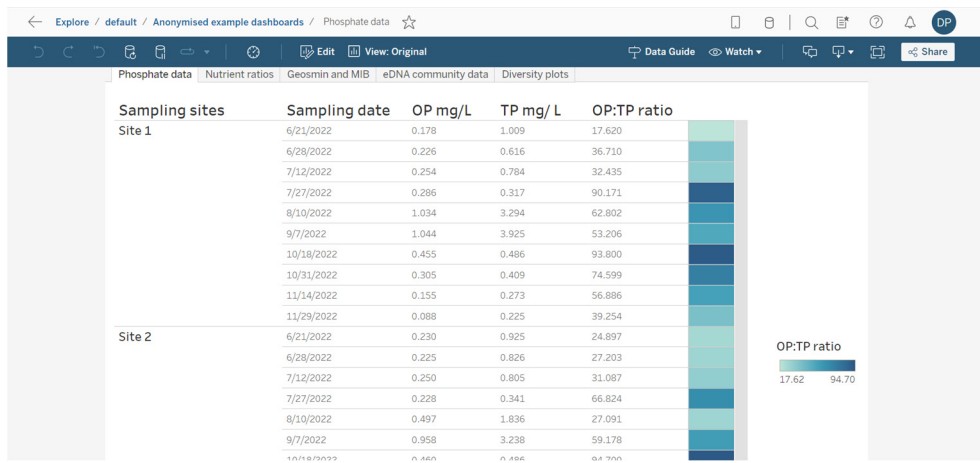

**Figure 6.** Example of how different dashboard data sheets can be presented as a series of data tabs that can easily be navigated once published on Tableau Cloud.

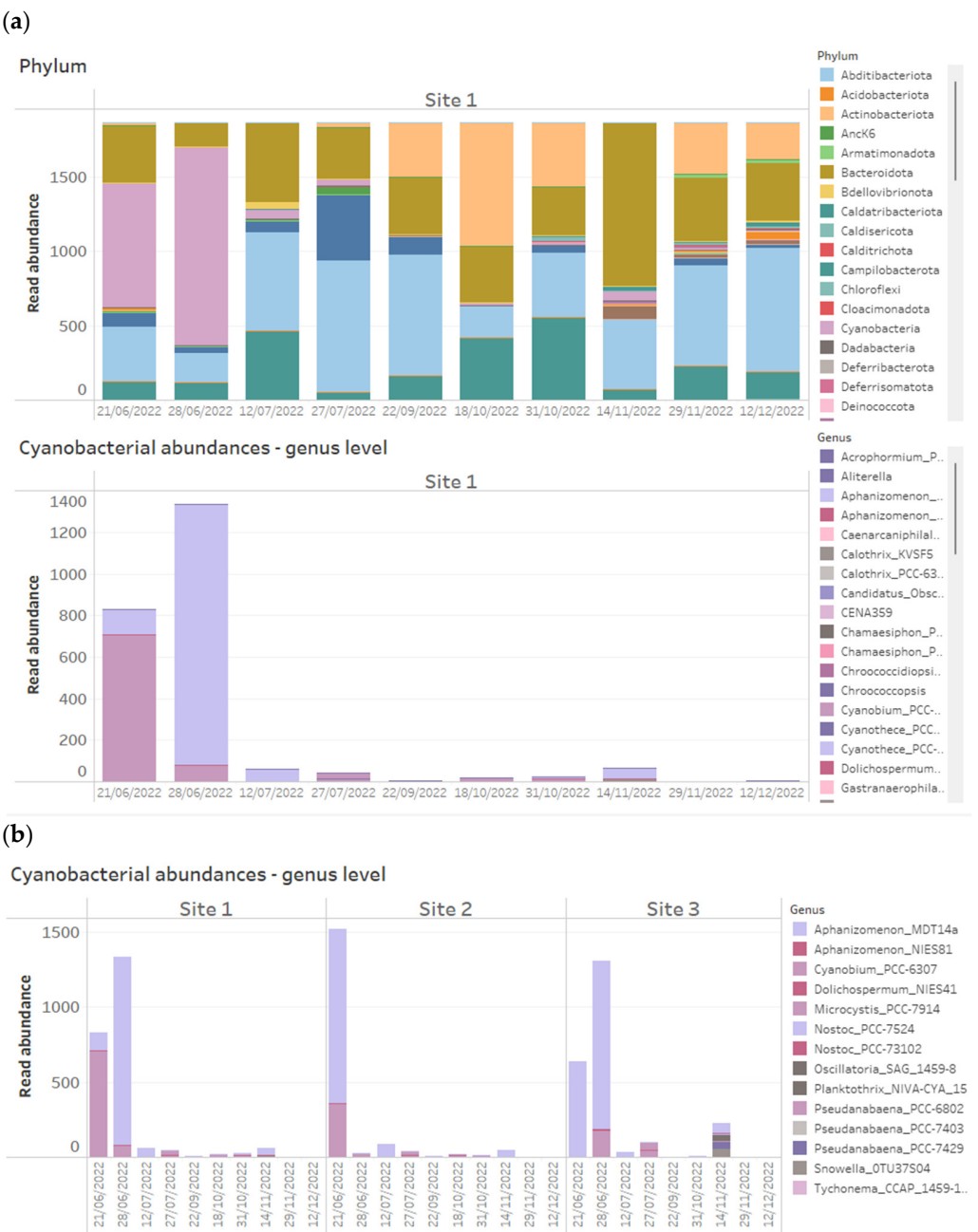

**Figure 7.** Example of how data can be easily filtered on Tableau Cloud based on (**a**) site or (**b**) taxa of interest (in this case known T&O producers) when presented as interactive dashboards. Here, we have filtered a number of water quality taxa belonging to the phylum, Cyanobacteria.

## 4. Conclusions

It is predicted that only a tiny portion of the Earth's water sources are made up of freshwater, and so freshwater supplies and potable water are fast becoming the most critical components of our future [1]. The sustainability of freshwater drinking supplies is threatened across the world by a plethora of different factors, including extreme weather events and the influx of anthropogenic-sourced pollution and nutrients [5]. Shifts in the way we utilize catchment areas for anthropogenic purposes put a strain not only on the quality of our water but also on the ways in which we can predict and manage those water quality events. Ultimately, implementing rapid, high-throughput methods for the identification of problem communities, as well as finding new and innovative methods for water management specialists to engage with modern technology, is critical in safeguarding

the sustainability of drinking water sources. Monitoring the sustainability of potable water sources requires sustainability-driven tools—tools which are developed directly in response to industry needs.

Here, we have demonstrated the benefit of a high-throughput sequencing approach to categorizing the bacterial communities, including Cyanobacteria, of reservoirs, a method which is able to provide extensive information about the reservoir communities as a whole and how they respond to a plethora of variables and stressors. Such a breadth of community taxa provides the ability to network data and hence use these data as biological indicators for sustainable water quality (providing a predictive capacity as well as a deeper understanding of reservoir processes). Further, we have shown the advantage of implementing an interactive dashboarding approach to ensure that what can be highly complex sequencing data are presented in such a way that is readily available for easy interpretation by reservoir management staff. Dashboards can be adapted and extended to make use of additional variables of interest that may arise in the future and can be personalized to water company needs. It is important to note that, in addition to Tableau, other platforms exist capable of manipulating and visualizing this type of data; here, we present just one of those options. UK water companies often outsource analysis for water quality determinants—we propose this outsourcing could be expanded to include eDNA community sampling and analysis or alternatively could be carried out in-house [38]. T&O events are expected to increase in their occurrence and magnitude globally in the future, so providing methods of accurate data generation, visualization and interpretation, as we suggest here, may become key to being able to pre-empt and manage the issue. The methods we outline here have been developed for the water industry in direct response to industry needs. Based on our methods, the generation of molecular data and its implementation into dashboards are currently being trialled and utilized by a number of water industry collaborators across the UK to continually monitor the quality and sustainability of their potable water sources and catchments. We propose that these methods can greatly aid reservoir management teams in their approach to T&O monitoring and can be used to implore more sustainable management pipelines, safeguarding future water sources.

**Author Contributions:** S.E.W. was responsible for manuscript concept and design, contributions to molecular laboratory work, bioinformatic and data analysis and manuscript writing and editing. C.H.T. was responsible for molecular laboratory work and manuscript editing. V.B. was responsible for water chemistry analysis and manuscript editing. T.R.B. and A.S.H. were responsible for intellectual input and manuscript editing. P.K. and R.G.P. were responsible for project concept, intellectual input, funding requisition and manuscript editing. All authors have read and agreed to the published version of the manuscript.

**Funding:** This study was kindly privately funded by a consortium of UK water industry partners; Ofwat, Bristol Water, Jersey Water, Scottish Water, United Utilities, Welsh Water and Yorkshire Water.

**Institutional Review Board Statement:** Not applicable.

**Informed Consent Statement:** Not applicable.

**Data Availability Statement:** The data presented in this study are available on request from the corresponding author. The data are not publicly available due to a non-disclosure agreement.

**Acknowledgments:** We would like to thank the Cardiff School of Biosciences Genome Research Hub and Biocomputing Hub for their technical support.

**Conflicts of Interest:** The authors declare no conflicts of interest.

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
