# Peer review of "Utilising eDNA Methods and Interactive Data Dashboards for Managing Sustainable Drinking Water"

_sustainability, doi:10.3390/su16052043_

Round 1

Reviewer 1 Report

Comments and Suggestions for Authors

Dear Authors

After a careful reading of this work, I believe that it is almost ready to be published.

The authors must write the aim and objectives of this work in page 3, line 115.

In my view, this work can be published after minor revision.

Author Response

Response to reviewer

Utilising eDNA methods and interactive data dashboards for managing sustainable drinking water

We thank you for taking the time to review our manuscript and for providing guidance on how to improve it. We have provided responses to individual comments below.

Reviewer 1

Comment 1:

Dear Authors

After a careful reading of this work, I believe that it is almost ready to be published.

The authors must write the aim and objectives of this work in page 3, line 115.

In my view, this work can be published after minor revision.

Response 1: We have now added this to line 110, where we say: “The aim of this manuscript is to facilitate the implementation of molecular data in reservoir management with the objective of presenting a dashboarding system that can be used to evidence base shifts in water quality risk.”

Reviewer 2 Report

Comments and Suggestions for Authors

The current manuscript provides a methodology to monitor taste and odor in drinking water reservoirs, utilizing a high-throughput sequencing approach. The writing is generally good, and the reviewer has the following comments:

1) The abstract could contain more details of the sequencing approach and reduce the number of generic descriptions.

2) The in-text citation format does not conform to MDPI requirements.

3) Lines 80-81: what is “me”hods’are” and “2tilized2d”?? The same happens at lines 106, 120, 142, 160, 196, 213... Please check the whole manuscript for similar incidents.

4) Line 146: Where is “step 4”?

5) Lines 189-211: If the current manuscript claims the increased speed and accuracy as the novelty of this study, why the current approach yields such benefits must be analyzed, examined, and proved. The methodology between sections 2.1-2.6 only provides generic descriptions of the approach but fails to prove that such an approach can yield the claimed benefits.

6) Lines 212-223: The “data dashboard” cannot be claimed as the novelty of this study since it is just a utilization of existing software.

7) All figures: the texts appear to be small and unclear. Please make sure that the submitted files are recognizable.

Comments on the Quality of English Language

English writing is OK

Author Response

Response to reviewer

Utilising eDNA methods and interactive data dashboards for managing sustainable drinking water

We thank you for taking the time to review our manuscript and for providing guidance on how to improve it. We have provided responses to individual comments below.

Reviewer 2

Comment 1:  The abstract could contain more details of the sequencing approach and reduce the number of generic descriptions.

Response 1: We thank you for this suggestion. However, as the journal appeals to a broad audience, we feel that the background information currently detailed within the abstract is both important and necessary in order to place our manuscript within the scope of Sustainability (as instructed by the Editor). Additionally, we feel that due to the word limit of the abstract, it would be difficult to introduce the details of sequencing technology in any more detail. We feel the details of the sequencing approach are best suited to the methods section.

Comment 2:  The in-text citation format does not conform to MDPI requirements.

Response 2: Thank you for pointing this out. This has now been amended throughout.

Comment 3:  Lines 80-81: what is “me”hods’are” and “2tilized2d”?? The same happens at lines 106, 120, 142, 160, 196, 213... Please check the whole manuscript for similar incidents.

Response 3: Thank you for pointing this out. We can see that this error is not present in the initial version we submitted and so it must have occurred somewhere during the submission process. We have now corrected this throughout the manuscript and will warn the Editor to look out for this in the next submission, in case it occurs again.

Comment 4:  Line 146: Where is “step 4”?

Response 4: This line refers to starting from ‘Step 4’ in the Dneasy Blood and Tissue Kit Protocol for Purification of Total DNA from Animal Blood or Cells. To clarify, we have amended this so that it now reads: “…starting from Step 4 in this commercially available protocol”.  

Comment 5:  Lines 189-211: If the current manuscript claims the increased speed and accuracy as the novelty of this study, why the current approach yields such benefits must be analyzed, examined, and proved. The methodology between sections 2.1-2.6 only provides generic descriptions of the approach but fails to prove that such an approach can yield the claimed benefits.

Response 5: Previous studies have demonstrated how the application of sequencing approaches for taxonomic identification are more accurate than traditional microscopy methods. Validating these methods was not the aim of this manuscript. The novelty within this manuscript is the entirely new application of how these methods can be implemented in regular water quality monitoring (alongside a novel dashboarding approach). We appreciate that the software is not novel but the application for presenting water quality risk level is new. What we present here is an example of how these data can be generated, presented and interpreted in a novel form for applied reservoir management.

Comment 6:  Lines 212-223: The “data dashboard” cannot be claimed as the novelty of this study since it is just a utilization of existing software.

Response 6: Please refer to our response above. Within our manuscript, we do not claim that the software is novel, instead we aim to highlight how useful it could be when used appropriately within water industry. Although the dashboarding software already exists, it is not currently utilized within water industry nor has it been used in conjunction with sequencing data to help simplify the complexity of this data type for non-expert use. As stated within the manuscript, we have trialed this approach with a number of UK industry partners who all have benefited from its implementation.

Comment 7:  All figures: the texts appear to be small and unclear. Please make sure that the submitted files are recognizable.

Response 7:  We will submit our figures separately to the journal in as high resolution possible, so that the journal can edit the formatting as they see fit. We were limited by the formatting requested by the journal regarding the initial submission. We will follow the same procedure as instructed previously and are happy to work with the Editor in the final print stages to maximise clarity as much as possible.

Reviewer 3 Report

Comments and Suggestions for Authors

The manuscript "Utilising eDNA methods and interactive data dashboards for managing sustainable drinking water" is devoted to the quality assesment of fresh water. The samples of water originated from three different places were subjected to genome sequencing to find out what microorganisms inhabit water in those places. In addition, orthophosphate and total phosphate concent was evaluated.

It goes without doubts that the short supply of fresh water is a major problem for the mankind, and all efforts in the field of water preservation, cleansing, or quality assesment are welcome. However, reading the present manuscript I felt like this is just a half of a story. Sure, Authors were able to show the diversity of microbial life in the water sources. But how this diversity is related to the quality of water? To taste and odour, more specifically? Why the hazardous blooming of algae were mentioned in the Intro section while only bacteria were identified in the actual report?

"We propose it is useful to monitor shifts in diversity as an early indicator system for broader ecological change, but diversity shifts (in either direction)
must be assessed within an ecological and community data context, paying close attention to various relevant environmental and chemical parameters" (lines 220-223). It is a promising suggestion indeed, but how exactly those shifts are linked with the parameters' variation? What parameters, exactly? For example, what have happened to water source 3 at 7 Sept 2022, which gave the observed peak in Shannon diversity? Was it for good or for ill?

I feel that the scope of the paper should be clarified taking into account the questions above.

Author Response

Response to reviewer

Utilising eDNA methods and interactive data dashboards for managing sustainable drinking water

We thank you for taking the time to review our manuscript and for providing guidance on how to improve it. We have provided responses to individual comments below.

Reviewer 3

The manuscript "Utilising eDNA methods and interactive data dashboards for managing sustainable drinking water" is devoted to the quality assesment of fresh water. The samples of water originated from three different places were subjected to genome sequencing to find out what microorganisms inhabit water in those places. In addition, orthophosphate and total phosphate concent was evaluated.

Comment 1: It goes without doubts that the short supply of fresh water is a major problem for the mankind, and all efforts in the field of water preservation, cleansing, or quality assesment are welcome. However, reading the present manuscript I felt like this is just a half of a story. Sure, Authors were able to show the diversity of microbial life in the water sources. But how this diversity is related to the quality of water? To taste and odour, more specifically? Why the hazardous blooming of algae were mentioned in the Intro section while only bacteria were identified in the actual report?

Response 1: Algal blooms are predominantly formed by blue-green algae, which are bacteria not green algae. We have made this clearer on line 45, where we now state:

“HABs, which predominantly consist of blue-green algae (correctly named as Cyanobacteria),…”

If we understand the reviewer correctly, they are indicating that there are fundamental links between community biodiversity and water quality risks. For example, the dominance of Cyanobacteria within harmful algal blooms. This is a well-studied aspect of limnology and to go in to details is beyond the scope of this paper. However, we would refer the reviewer to Hooper et al. (2023), which is cited within the manuscript, as an example of how this effects taste and odour risk. We also refer the reviewer to line 214 where we address the importance of this and hence why this novel use of molecular data is of high benefit to water reservoir managers.

Comment 2: "We propose it is useful to monitor shifts in diversity as an early indicator system for broader ecological change, but diversity shifts (in either direction) must be assessed within an ecological and community data context, paying close attention to various relevant environmental and chemical parameters" (lines 220-223).

It is a promising suggestion indeed, but how exactly those shifts are linked with the parameters' variation? What parameters, exactly? For example, what have happened to water source 3 at 7 Sept 2022, which gave the observed peak in Shannon diversity? Was it for good or for ill?

Response 2: Regarding your comment on diversity, please refer to our answer above. How diversity is related to water quality varies based on the ecology of the area and should be interpreted accordingly. As such, a change in diversity can be beneficial to increase resilience to change or alternatively facilitate a dominance of harmful taxa that can result in an increased level of water quality risk such as taste and odour metabolite production by Cyanobacteria. We have added a line to this effect within the manuscript at line 217.

Round 2

Reviewer 2 Report

Comments and Suggestions for Authors

Thanks for correcting the formatting issues of the manuscript. However, the manuscript still suffers from novelty issues. As the authors agreed, the methodology is not novel. The only "novelty" (if any) is utilizing a software dashboard in the water industry. Purely utilizing an existing technology does not satisfy the requirements of academic journals. The reviewer recommends submitting the current manuscript to trade journals instead.

Comments on the Quality of English Language

English is OK.

Author Response

Report form comment 1: Is the content succinctly described and contextualized with respect to previous and present theoretical background and empirical research (if applicable) on the topic?

Response: We believe this has now been addressed in our main response detailed below. 

Report form comment 2: Are the research design, questions, hypotheses and methods clearly stated?

Response: Our aims and objectives were added to the introduction in the previous version of amendments (lines 114-116). Please provide clarification if further information is required.

Report form comment 3: Are the arguments and discussion of findings coherent, balanced and compelling?

Response: We assume from your comment below, that this refers to the ‘compelling’ component of this statement. Please see our response below for why we feel the work is in fact compelling.

Comment 1: Thanks for correcting the formatting issues of the manuscript. However, the manuscript still suffers from novelty issues. As the authors agreed, the methodology is not novel. The only "novelty" (if any) is utilizing a software dashboard in the water industry. Purely utilizing an existing technology does not satisfy the requirements of academic journals. The reviewer recommends submitting the current manuscript to trade journals instead.

Response 1:

Thank you for your commentary and for taking the time to consider our work. To address your comment, while eDNA is not a novel concept in itself (as we state in lines 101-103), the application of eDNA methodology within water quality monitoring is still in its early stages– we work with a number of UK and international water industry partners, none of which have used these methods for water quality monitoring and none of which were aware of their application to drinking water management until our research team provided that outlet. Bridging the gap between research and real-world need is an incredibly important part of our roles as scientists and is only achieved through science communication and published evidence of concepts. We have it on good authority from a number of water industry partners, that without published proof that these concepts are achievable, they are unable to acquire permission to implement those methods in to standard practice. Although the laboratory methods are used within a research context, in our manuscript we present optimized methods for reservoir samples and therefore provide a resource for water industry looking to implement these methods, while the dashboarding concept detailed within the manuscript enables the integration of multiple data types of choice, taking what is incredibly complex bioinformatic data and presenting it in a way that is useful to reservoir management teams. Communicating such a huge plethora of data in an industry relevant format is challenging – hence here we look to establish methods by which this can be achieved through this publication. We know what we present here is critical to water industry, and provides compelling applied methods in direct response to industry need, because we work closely with a number of them and are constantly approached by new partners looking to adopt the methods but not knowing how. We have now made this clearer in lines 112-114. We believe that method optimisation, data integration and communication provides significant novelty within our manuscript.

Given the context of the special issue in which we hope to publish this manuscript (i.e. “Innovations in Water Quality Improvement Technologies: Current Advances and Future Directions”), we feel this work is a perfect match. We clearly demonstrate the methods currently available and provide an important contribution that demonstrates the future direction in which these novel innovations can bring research tools to an area benefitting industry.

Reviewer 3 Report

Comments and Suggestions for Authors

The clarification provided by the Authors is welcome. I guess, now the manuscript is in satisfactory shape for publication.

Author Response

Comment: The clarification provided by the Authors is welcome. I guess, now the manuscript is in satisfactory shape for publication.

Response: Thank you. And thank you for your time throughout this review process.

Round 3

Reviewer 2 Report

Comments and Suggestions for Authors

Thanks for the explanation. The reviewer agrees with the authors' opinions.

Comments on the Quality of English Language

No specific issues.